# LSTT: Long Short-Term Transformer for Video Small Object Detection

## Abstract

Detecting small objects in video sequences is crucial, yet it poses significant challenges due to their limited visibility and dynamic nature, which complicates accurate identification and localization. Traditional methods often employ a uniform aggregation strategy across all frames, neglecting the unique spatiotemporal relationships of small objects, which results in insufficient feature extraction and diminished detection performance. This paper introduces a long short-term transformer network specifically designed for small object detection in videos. The model integrates features from both long-term and short-term frames: long-term frames capture global contextual information, enhancing the model's ability to represent background scenes, while short-term frames provide dynamic information closely related to the current detection frame, thereby improving the feature representation of small objects. A dynamic query generation module optimizes query generation based on the implicit motion relationships of targets in short-term frames, adapting to the current video framework. Additionally, the network employs a progressive sampling strategy—densely sampling short-term frames and sparsely sampling long-term frames—to effectively model video scenes. A spatio-temporal alignment encoder further enhances pixel-level features by accounting for temporal and spatial transformations. Extensive experiments on the VisDrone-VID and UAVDT datasets demonstrate the method's effectiveness, with an average detection precision increase of 1.4% and 2.1%, respectively, highlighting its potential in small object video detection.

## 1 Introduction

Video small object detection refers to the task of identifying and localizing objects of small size within video sequences, which are often characterized by limited pixel representation and subtle visual features. This process is essential in various fields, including self-driving cars, satellite imagery, healthcare imaging, and industrial quality control (Jiao et al., 2022; Wang et al., 2020; Jiang et al., 2022). The challenge lies in the inherent difficulties associated with small objects, such as their low resolution and tendency to blend into complex backgrounds. As a result, research has increasingly focused on enhancing feature detection for small objects (Lim et al., 2021a; Feng et al., 2020b; Ashraf et al., 2021b; Zhang et al., 2021). Studies have demonstrated that utilizing temporal information across video frames is crucial for improving detection performance, as single-frame detection often struggles with issues like motion blur, low resolution, and the diminutive size of the objects. Leveraging information from multiple frames helps to overcome these challenges (Bertasius et al., 2018a; Han et al., 2020; Cui et al., 2021; Ma et al., 2022).

Given the strong visual similarity and temporal continuity between adjacent frames, previous methods have leveraged temporal information through pixel-level enhancements across these frames(Zhu et al., 2017; Bertasius et al., 2018b). Small objects, which may only occupy a few pixels in a single frame, are particularly susceptible to pixel-level noise and environmental changes, making pixel-level feature enhancement crucial(Xiao et al., 2023b). These methods typically employ deformable convolutions or optical flow to align and aggregate features between adjacent frames. Although pixel-level alignment has successfully enhanced features, it primarily focuses on single-pixel alignment and aggregation, neglecting the aggregation of features in adjacent pixel areas. Additionally, some studies(Zhou et al., 2022) have attempted to aggregate features in neighboring regions of corresponding pixels across frames using deformable attention mechanisms. However, they have not

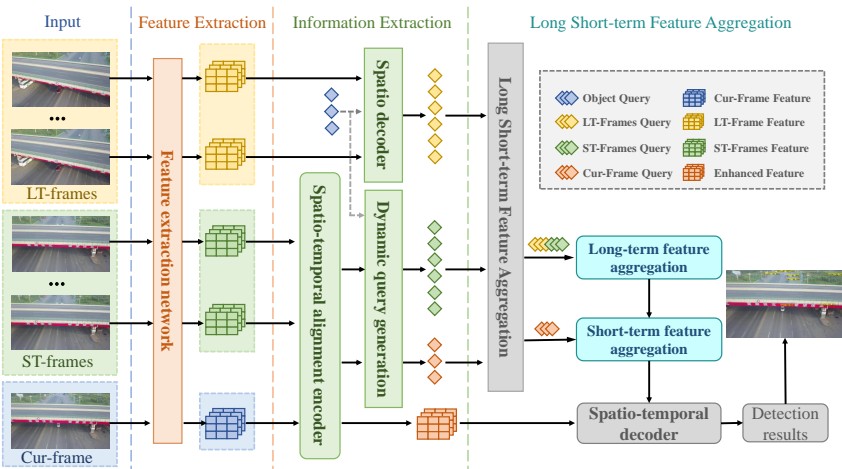

Figure 1: The architecture of the proposed LSTT. The current frame (Cur-frame), short-term frames (ST-frames) near the Cur-frame, and long-term frames (LT-frames) sampled from the whole video first go through the feature extraction network. Initially, these frames pass through a feature extraction network. Subsequently, the features of the Cur-frame and the ST-frames are integrated using a spatio-temporal feature alignment encoder. Long-term global information from LT-frames is extracted through a spatial decoder, while dynamic querying of ST-frames and the Cur-frame produces target queries that better conform to the current video sequence. Finally, features are aggregated over multiple layers, depending on whether positional encoding information is added, to effectively combine long-term and short-term features. Best viewed in color and zoomed in.

achieved precise pixel-level feature alignment, overlooking spatial transformations between adjacent frames.

The aggregation of temporal information from adjacent frames is often limited to a brief time window. To fully exploit the spatio-temporal information across an entire video, recent methods(Xiao et al., 2023b) have focused on the roles of long-term and short-term frames in detection. Some approaches(He et al., 2021; Zhu et al., 2017; Bertasius et al., 2018b) use optical flow and deformable convolutions to extract and aggregate short-term frame features; however, these are challenging to train and costly. Other methods focus on the semantic information of long-term frames to establish long-term dependencies(Wu et al., 2019; Fujitake & Sugimoto, 2022), often involving randomly sampling frames from the video, which leads to unstable detection results and information loss. Notably, some strategies(Chen et al., 2020) define long-term and short-term frames based on their temporal distance, using attention mechanisms to aggregate features over time and space throughout the video. However, whether aggregating long-term or short-term frames, such methods typically employ a generic approach that fails to consider the varied information within frames concerning small objects, resulting in suboptimal detection performance. Previous research has shown that contextual information is crucial for detecting small objects(Lim et al., 2021b; Xiao et al., 2023a). The aggregation of video frames frequently lacks attention to global contextual information, particularly for short-term frames, which tend to contain similar global context. In contrast, long-term frames, while rich in global context, often lack effective extraction of this information.

Despite the establishment of end-to-end networks by video object detectors utilizing temporal information, their performance in detecting small-sized targets is typically suboptimal(Zhou et al., 2022). Video small object detectors often rely on complex post-processing steps, such as establishing tubelet links among targets in video frames. Consequently, there is an urgent need for a more efficient end-to-end video small object detector.

To address these challenges, this paper proposes an end-to-end long short-term transformer (LSTT) network. Compared to previous methods, this network avoids complex post-processing steps while fully accounting for the information discrepancies between different video frames. It efficiently extracts and aggregates information from both long-term and short-term frames, leveraging the transformer architecture to enhance the detection of small objects. Given the visual similarity between

adjacent frames, we designed a spatio-temporal alignment encoder to mitigate the effects of spatial transformations between frames. This encoder uses deformable attention to sparsely sample and aggregate features in neighboring regions of corresponding pixels across frames. Recognizing that different frames contain diverse information, as illustrated in Figure 1, we define video frames based on their inherent connections rather than their temporal proximity. Long-term frames provide scene understanding, while short-term frames capture the motion and appearance characteristics of objects. For long-term frame sampling, we use a progressive strategy: densely sampling short-term frames and sparsely sampling long-term frames. We developed a feature aggregation module that extracts global contextual information from long-term frames using a deformable attention decoder, while short-term frames utilize positional encoding and texture features to aggregate motion and appearance information. To address the uneven distribution of small targets that can lead to positional encoding errors, we designed a dynamic query generation method to obtain accurate positional information. The system differentiates queries from long-term frames containing scene information, short-term frames containing location and appearance details, and current frames, aggregating these to effectively extract information across the entire video. Through these targeted efforts, our LSTT model outperforms existing methods on the Visdrones2019-VID and UAVDT datasets.

Our main contributions are summarized as follows:

- An end-to-end video small object detection network, the long short-term transformer , is proposed, which significantly enhances feature representation and detection accuracy for small objects.

- A spatio-temporal alignment encoder is introduced, effectively mitigating the impact of spatial transformations between frames by using deformable attention mechanisms to achieve precise feature alignment and aggregation.

- A novel sampling strategy is developed, dynamically balancing the extraction of global contextual information from long-term frames and detailed motion and appearance features from short-term frames, ensuring comprehensive temporal coverage in videos.

- A long short-term feature aggregation module is designed, combining features from various frames to effectively aggregate global contextual information from long-term frames and motion and appearance information from short-term frames. This module utilizes a dynamic query generation method to generate precise positional encodings, which, combined with appearance features, aggregate short-term frames, and a deformable attention decoder to extract and aggregate global information from long-term frames.

## 2 RELATED WORKS

Object detection from images has seen significant advancements with the introduction of several leading detectors in recent years. Following the development of early detectors, video object detection has been extensively studied as a more challenging task. Results indicate that leveraging information from other frames can significantly improve detection in the current frame. Therefore, temporal feature aggregation has gained increasing attention and is now a key component of video small object detection.

**Temporal feature aggregation:** Video object detection necessitates the propagation of temporal information across frames, typically achieved by aggregating features from adjacent frames to enhance the detection frame's feature representation. Early methods employed optical flow warping for feature aggregation(Zhu et al., 2017; Bertasius et al., 2018b). The DFF(Zhu et al., 2018) method utilizes optical flow networks to predict flow fields, aligning keyframe features with short-term frames to reduce redundant computations and accelerate the network. FGFA(Zhu et al., 2017) employs optical flow to align and fuse short-term frame features with the current frame, enhancing detection accuracy. MANet(Wang et al., 2018) builds upon FGFA by implementing pixel and instance-level feature alignment and aggregation, and employs a motion pattern reasoning module for further feature aggregation. Some methods expand the temporal window by integrating long-term frame features with the detection frame to establish long-range dependencies. SELSA(Wu et al., 2019) enhances feature aggregation by calculating semantic similarity between the current and long-term frames. STMN(Xiao & Lee, 2018) employs recurrent computation units as spatiotemporal memory modules to convey semantic information across frames. MEGA(Chen et al., 2020) incorporates both

global and local information from videos, utilizing long-term memory for feature enhancement. TF-Blender(Cui et al., 2021), in contrast, employs a learnable network to predict aggregation weights, unlike methods relying on cosine similarity.

Most approaches are based on the classical two-stage Faster R-CNN network(Ren et al., 2015), which involves complex post-processing steps. Recently, Transformer-based object detection networks have optimized the process, enabling end-to-end detection in single-frame images with competitive performance. However, these methods face temporal and spatial limitations when applied to videos. TransVOD, TransVODLite, and TransVOD++(Zhou et al., 2022) incorporate spatiotemporal decoders and query generation across multiple video frames, balancing speed and accuracy. PTSEFormer(Wang et al., 2022) employs a progressive strategy involving multi-scale feature extraction, focusing on temporal information and spatial transitions between frames. Sparse video object detection employs an end-to-end trainable detector leveraging temporal information to propose region suggestions. Conversely, DAFA(Roh & Chung, 2022) emphasizes global over local temporal features of videos. DEFA critiques FIFO memory inefficiencies, proposing a diversity-aware memory for instance-level feature retention in attention modules rather than frame-level memory. VS-TAM(Fujitake & Sugimoto, 2022) improves feature quality element-wise through sparse aggregation before detecting object candidate regions, and utilizes external memory for long-term contextual information.

These methods propagate temporal information through feature aggregation, performing well on regular objects but failing to account for intrinsic connections between video frames. They use uniform aggregation strategies across the entire temporal window of a video, struggling to extract key information beneficial for small object detection, thus limiting performance improvements in this area.

**Video small object detection:** Research indicates that video analysis can enhance small object detection performance by leveraging temporal continuity. For instance, the Motion R-CNN(Feng et al., 2020a) and DogFight(Ashraf et al., 2021a) algorithms analyze dynamic changes in video sequences, extracting crucial information about object movement and changes, thus enhancing the accuracy of small object detection. Algorithms like STDnet-ST++(Bosquet et al., 2021) and FANet(Cores et al., 2023) demonstrate how constructing and optimizing spatiotemporal trajectories improve small object recognition. These algorithms associate detection results across consecutive frames, building trajectories and further refining them to eliminate noise and improve detection signals. The LSTFE(Xiao et al., 2023b) method extends this concept by fusing long-term and short-term video features, aggregating diverse information from different frames, significantly improving small object recognition capabilities and demonstrating the effectiveness of long short-term structures. However, these two-stage network-based video small object methods often require manual refinement of complex post-processing steps, impeding the realization of end-to-end object detection.

Although both short-term and long-term frame features have been used to enhance detection performance, the information extracted from these features remains insufficient to establish stable connections among small objects. LSTFE(Xiao et al., 2023b) indicated that distinguishing between long-term and short-term frames in aggregation provides a clear advantage for small object features. However, convolutional neural network-based feature extraction struggles to capture sufficient environmental information crucial for small objects. Considering the global context modeling capability of transformers, we applied a transformer-based detector to video small object detection for the first time. Specifically, we considered both temporal and spatial information, performing sparse sampling aggregation among short-term frames rather than simply aggregating weights between adjacent frames like TransVOD(Zhou et al., 2022). Moreover, unlike the aforementioned methods focusing on either short-term or long-term frame information, we defined video frames from the perspective of intrinsic connections, fully considering the different information between short-term and long-term frames, and aggregating object queries accordingly. By adopting these designs, our approach is more conducive to detecting small objects in videos compared to previous works.

## 3 METHODS

### 3.1 FRAMEWORK OVERVIEW

The architecture of LSTT, shown in Fig. 1, processes the current frame along with multiple short-term and long-term frames sampled from the video. The feature extraction network first extracts features from these frames. Short-term frames, adjacent to the current frame, share similar appearance information. The spatio-temporal alignment encoder uses convolutional offsets to align features between the current and short-term frames. Additionally, a deformable attention mechanism sparsely samples effective features from short-term frames, enhancing the pixel-level features of the current frame.

For long-term frames, direct feature alignment or pixel-level fusion is impractical due to lack of temporal continuity. Instead, global semantic information is extracted from long-term frames using object queries and fused with short-term frame queries. Positional encoding represents the relative positions of objects in short-term frames, while dynamic query generation learns relative motion relationships, addressing the uneven distribution of small objects. Progressive random sampling with exponential segmentation ensures coherent motion relationships in short-term frames and adequate scene information from long-term frames.

### 3.2 PROGRESSIVE RANDOM SAMPLING STRATEGY

Short-term frames exhibit similar global contexts within a brief temporal window. To capture key scene changes, we sample long-term frames from the entire video. Traditional sampling methods lack sensitivity to temporal changes and fail to capture scene information effectively.

We propose a progressive random sampling strategy with two core aspects. First, exponentially divide the video timeline, ensuring higher sampling density for segments near the current frame to capture short-term dynamics.

For distant video segments, we apply sparser sampling to expand the temporal window and capture scene-rich long-term frames. Random sampling within each segment adds diversity. Given video frames $\{F_t\}_{t=1}^{T} \in \mathbb{R}^{H_0 \times W_0 \times C_0}$, where $T$ is the video length and $H_0$, $W_0$, and $C_0$ are frame dimensions, we sample $k$ reference frames—densely for short-term and sparsely for long-term frames. This is defined as $F_{\text{space}} = \{2^i \mid 0 \leq i < k\}$, selecting $m$ long-term frames $F_g$ and $n$ short-term frames $F_s$.

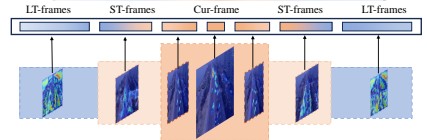

Figure 2: Process of progressive random sampling strategy.

The reference frames include densely packed short-term frames within an extremely short time window and long-term frames that contain extensive scene transition information. Compared to commonly used uniform sampling, our method captures more concentrated short-term motion information and broader scene data with the same number of reference frames.

### 3.3 SPATIO-TEMPORAL ALIGNMENT ENCODER

In video object detection, the current frame $F_c$ and short-term frames $F_s$ often exhibit spatial misalignment due to motion, degrading feature fusion quality. The STAE employs deformable multi-head attention to selectively sample and align features, enhancing pixel-level feature representation.

The encoder first extracts spatial features from both $F_c$ and $F_s$, producing tensors $M_c$ and $M_s$. These tensors, each with dimensions $[C, H, W]$, are concatenated to form $M_{\text{cat}}$. This concatenated feature map passes through a convolutional layer to compute motion offsets $m_o$, which are crucial for aligning features across the frames.

To effectively integrate features from multiple short-term frames, the encoder uses multi-head deformable attention. This process is described by the equation:

$$M'_{t+s} = \sum_{l=1}^{L} \sum_{k=1}^{K} A_{lqk} \cdot W' x^l (\phi_l(\hat{p}_q) + \rho(\hat{p}_{lq}) + \Delta p_{lqk}) \tag{1}$$

Here, $W'$ represents the mapping to value vectors, $\phi_l(\hat{p}_q)$ denotes the normalized sampling reference point within the feature map coordinates, $\rho(\hat{p}_{lq})$ adjusts for spatial offsets in the $l$-th frame, and $\Delta p_{lqk}$ specifies the displacement for the $k$-th sampling point in the $l$-th frame. This selective sampling mechanism significantly enhances the integration of temporal and spatial information, optimizing feature aggregation at a pixel level for improved detection accuracy.

This mechanism allows the STAE to dynamically adjust both spatially and temporally by learning the necessary offsets to align and effectively aggregate features from the short-term frames, thereby enhancing the pixel-level feature representation of the current frame.

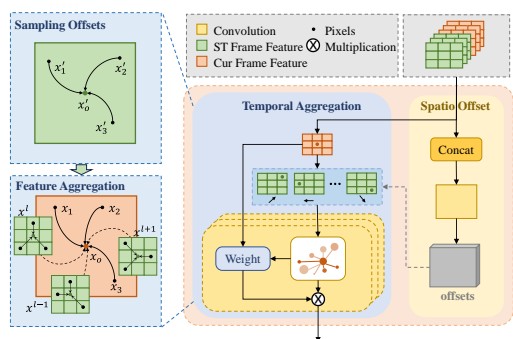

Figure 3: Process of spatio-temporal alignment encoder.

### 3.4 LONG SHORT-TERM FEATURE AGGREGATION

Short-term frames enhance the current frame within a local temporal range, but lack sufficient global context. Long-term frames provide diverse object features and global context crucial for small object detection. By extending the temporal window, the model integrates scene changes and uses long-term context for comprehensive video modeling.

The long short-term feature aggregation (LSTFA) module aggregates object queries from long-term and short-term frames into the current frame's queries. LSTFA includes long-term and short-term feature aggregation stages.

**Long-term feature aggregation:** A weight-sharing feature extraction network obtains image features $M_g$ and query vectors $Q_t$ from long-term frames $F_g$. These inputs feed into a spatial decoder, generating long-term frame object queries $Q_g = \{q_1^g, q_2^g, \ldots, q_n^g\}$. Short-term frame query vectors $Q_s = \{q_1^s, q_2^s, \ldots, q_m^s\}$ aggregate with long-term frame queries through multi-head attention, resulting in enhanced query features $Q'_s = \{q_1^{s'}, q_2^{s'}, \ldots, q_n^{s'}\}$.

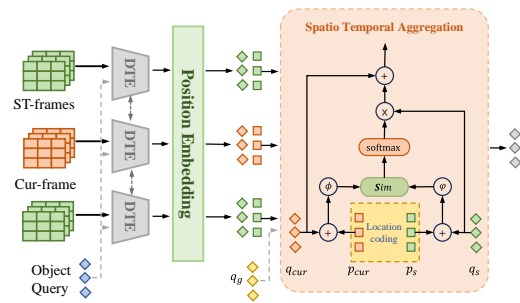

Figure 4: Process of long short-term feature aggregation.

Queries and keys for long-term and short-term frame vectors are generated using fully connected layers. Attention weights based on appearance similarity are calculated:

$$s_{i,j}^h = \text{softmax}(\text{Sim}(\phi_h(q_i^s), \varphi_h(q_j^g))) \tag{2}$$

Similarity function Sim can be formulated as:

$$Sim(X, Y) = \frac{X \cdot Y}{\sqrt{dim}} \tag{3}$$

Outputs are aggregated and adjusted through attention head weights $W_h$ and transformation matrix $W_p^h$:

$$q_i^{s'} = q_i^s + \sum_{h=1}^{H} W_h \sum_{j=1}^{n} s_{i,j}^h (W_p^h \cdot q_j^g), \quad i = 1, \cdots, m \qquad (4)$$

The overall long-term feature aggregation function is:

$$Q_s' = \zeta_g(Q_s, Q_g) \qquad (5)$$

**Short-term feature aggregation:** Short-term frames maintain spatial association with the current frame. Positional encoding models this continuity, and dynamic queries address uneven distribution. Inputs for aggregation are the query feature vectors $Q_{cur}$ and positional encoding $P_{cur}$ of the current frame, and enhanced short-term frame query vectors $Q_s'$ and positional encoding $P_s$:

$$P_s, Q_s = \{G(Q_s, M_i) | i(1, \cdots, n)\} \qquad (6)$$

Multi-head attention and self-attention achieve interactions between targets in the current and short-term frames:

$$s_{i,j}^h = \text{softmax}(\text{Sim}(\phi_h(q_i^{cur} + p_i^{cur}), \varphi_h(q_j^s + p_j^s))) \qquad (7)$$

The overall short-term feature aggregation function is:

$$Q_{cur}' = \zeta_s(Q_{cur}, Q_s) \qquad (8)$$

Through the STAE and LSTFA modules, we obtain enhanced current frame features $M_{t+s}'$ and query vectors $Q_{cur}'$. The deformable attention decoder processes these features, followed by a neural network for classification and bounding box regression, yielding the detection results.

# 4 EXPERIMENTS

## 4.1 EXPERIMENT SETUP

**Datasets:**For a fair and convincing comparison, we chose to conduct extensive experiments on two publicly available challenging datasets:

- *VisDrone2019-VID dataset*:This dataset includes 96 challenging HD video sequences from UAV perspectives, with 56 sequences for training (24,198 frames), 7 for validation (2,648 frames), 16 for challenge testing (6,322 frames), and 17 for development testing (6,635 frames). It covers 10 object categories: pedestrians, people, cars, vans, buses, trucks, motorcycles, bicycles, awning tricycles, and tricycles.

- *UAVDT dataset*:Designed for vehicle detection, this dataset comprises 50 videos captured by drone-mounted cameras, totaling around 40,000 annotated frames. It includes 30 videos for training and 20 for testing, with resolutions of 1080x540 pixels. The test set contains 375,884 objects across three categories: cars (361,055), trucks (7,595), and buses (7,234), with 20.3% of the objects being small.

**Implementation Details:** Our model was trained and tested on four 24GB NVIDIA RTX 4090 GPUs, using ResNet101 and ResNet50 as backbones, and Deformable DETR as the base detector. We employed ImageNet pre-trained weights, training for 10 epochs on images followed by 7 fine-tuning epochs on videos. The input sizes were 1024x540 pixels for UAVDT and 1920x1080 pixels for VisDrone2019-VID. Training used a learning rate of 0.0002 with AdamW optimizer, processing one image per GPU batch. For reference frames, we sampled eight long-term and six short-term frames relative to the current frame. The same input sizes were used during inference, and the batch size was kept at 1 to simplify the detection pipeline for Video Small Object Detection (VSOD).

| Algorithm | Backbone | $mAP$ | $AP_s^{@.5}$ |
|---|---|---|---|
| Faster R-CNN(Ren et al., 2015) | ResNet-101 | 31.80 | 17.60 |
| FPN(Lin et al., 2017) | ResNet-101 | 37.12 | 19.72 |
| DFF(Zhu et al., 2018) | ResNet-101 | 33.16 | 16.80 |
| FGFA(Zhu et al., 2017) | ResNet-101 | 35.26 | 17.40 |
| RDN(Deng et al., 2019) | ResNet-101 | 37.03 | 18.67 |
| SELSA(Wu et al., 2019) | ResNet-101 | 37.56 | 19.53 |
| MEGA(Chen et al., 2020) | ResNet-101 | 39.33 | 20.56 |
| LSTFE(Xiao et al., 2023b) | ResNet-101 | 41.86 | 21.79 |
| Single Frame Baseline(Zhu et al., 2020) | ResNet-101 | 33.62 | 17.90 |
| TransVOD(Zhou et al., 2022) | ResNet-101 | 39.69 | 19.62 |
| Proposed | ResNet-101 | 44.26 | 23.19 |

Table 1: Comparison results on VisDrones2019-VID test dev (%)

| Algorithm | Backbone | $AP_s^{@.5}$ |
|---|---|---|
| Faster R-CNN(Ren et al., 2015) | ResNet-101 | 28.6 |
| FPN(Lin et al., 2017) | ResNet-101 | 31.2 |
| FGFA(Zhu et al., 2017) | ResNet-101 | 24.8 |
| RDN(Deng et al., 2019) | ResNet-101 | 28.5 |
| MEGA(Chen et al., 2020) | ResNet-101 | 31.8 |
| LSTFE(Xiao et al., 2023b) | ResNet-101 | 34.4 |
| Single Frame Baseline(Zhu et al., 2020) | ResNet-101 | 29.6 |
| TransVOD(Zhou et al., 2022) | ResNet-101 | 30.8 |
| Proposed | ResNet-101 | 36.5 |

Table 2: Comparison results on the UAVDT dataset (%)

## 4.2 Comparison with state-of-the-art

The proposed algorithm is compared with state-of-the-art methods on the UAVDT and VisDrone2019-VID datasets to verify its effectiveness. The overall detection performance of each algorithm model is assessed using $mAP$, while $AP_s^{@.5}$ is used to evaluate the detection performance of each algorithm model on small objects within the datasets.

**VisDrones2019-VID test dev:** The comparative results of video object detection algorithms on the VisDrones2019-VID dataset are detailed in Table 3.

Table 3 presents the comparative performance of video object detection algorithms on the VisDrones2019-VID dataset. The proposed algorithm significantly leads with an $mAP$ of 44.26% and $AP_s^{@.5}$ of 23.19%, showing improvements of 10.64% and 5.29% respectively over the baseline. FGFA and DFF, using optical flow, show limited gains in $mAP$ (up to 1.64%) and marginal or no improvements in $AP_s^{@.5}$. The FPN network achieves a 3.50% and 1.82% boost in $mAP$ and $AP_s^{@.5}$, benefiting from multi-scale feature fusion. MEGA, utilizing attention mechanisms, improves by 5.71% and 2.66% in both metrics, demonstrating effective long-term dependency handling. LSTFE, specialized for small object detection, marks the highest rise among two-stage networks with increases of 10.06% in $mAP$ and 4.19% in $AP_s^{@.5}$. In contrast, TransVOD, leveraging temporal information with a transformer architecture, shows a notable $mAP$ enhancement of 6.07% but only a slight increase in $AP_s^{@.5}$ (0.72%). Overall, the proposed algorithm outshines others in effectively detecting both general and small targets, underscoring its superior design and performance.Fig. 5 shows some qualitative comparison results of our LSTT versus other state-of-the-art works.

**UAVDT dataset:** This dataset is used for video object tracking and detection, with only a few image detectors reported for the $AP_{75}$ metric on this dataset. To ensure a fair comparison, this chapter replicates the performance of classical algorithms on this dataset, including the optical flow-based FGFA algorithm, attention mechanism-based MEGA algorithm, long short-term feature aggregation-based LSTFE algorithm, and Deformable DETR-based TransVOD algorithm. Detailed comparative results of video object detection algorithms on the UAVDT dataset are reported in Table 2.

In the UAVDT dataset, our LSTT algorithm stands out with a top $AP_s^{@.5}$ of 36.5%, demonstrating superior small object detection capabilities in dynamic video environments. LSTFE also performs impressively, securing an $AP_s^{@.5}$ of 34.4% by effectively leveraging long short-term feature dy-

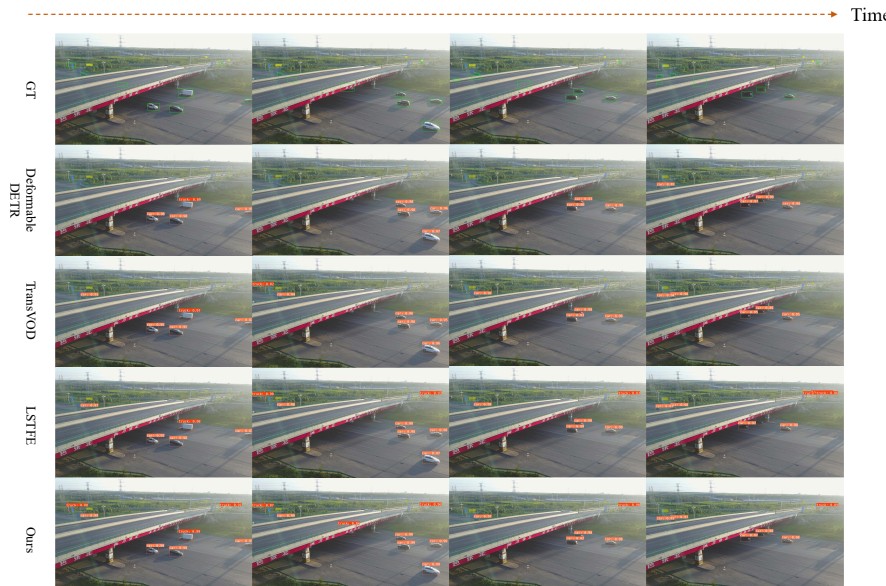

Figure 5: Visualized comparison against the state-of-the-art methods on Visdrones-VID dataset. We show the detection result of our method against Deformable DETR, TransVOD and LSTFE. All models employ ResNet-101 as the backbone.

| Algorithm | LT Frames | ST Frames | $AP_s^{@.5}$ |
|---|---|---|---|
| Uniform Sampling | | ✓ | 21.88 |
| Random Sampling | | ✓ | 21.63 |
| Progressive Random Sampling | | ✓ | 22.17 |
| Uniform Sampling | ✓ | ✓ | 22.36 |
| Random Sampling | ✓ | ✓ | 22.52 |
| Progressive Random Sampling | ✓ | ✓ | 23.19 |

Table 3: Ablation study of sampling strategy (%)

namics for enhanced small target accuracy. MEGA and FPN, with $AP_s^{@.5}$ scores of 31.8% and 31.2% respectively, show solid performance through attention mechanisms and multi-scale feature fusion, which are crucial for detecting small objects across varying resolutions. TransVOD, achieving 30.8%, benefits from its transformer-based architecture, offering substantial improvements in handling dynamic scenes, albeit slightly below the top performers. In contrast, FGFA, which depends heavily on optical flow, trails with an $AP_s^{@.5}$ of 24.8%, underscoring its limitations in tracking and detecting small, fast-moving objects.

## 4.3 ABLATION STUDIES

We conduct ablation studies to validate the effectiveness of each component in our model on the Visdrones2019-VID dataset.

**Progressive random sampling strategy** To assess the efficacy of sampling strategies, we conducted comparative experiments on uniform sampling, random sampling, and the progressive random sampling proposed in this study, evaluating each strategy's effectiveness based on the inclusion of long-term and short-term frames. The results are detailed in Table 3, which presents an ablation study of these sampling strategies. This table illustrates the impacts on the $AP_s^{@.5}$ metric when either long-term or short-term frames are added across different algorithms and sampling methods.

Uniform sampling, when only short-term frames are included, performs slightly worse than the strategy employed in this paper. This is because while it maintains consistent time intervals, it lacks the flexibility needed to capture tightly packed motion information effectively. In contrast, random sampling performs the worst under the same conditions, likely because its randomness complicates the extraction of effective motion information from these frames. However, when long-term frames

| Algorithm | Backbone | $AP_s^{@.5}$ |
|---|---|---|
| without spatial offsets | ResNet101 | 22.68 |
| with spatial offsets | ResNet101 | 23.19 |
| ST-Frame with positional encoding | ResNet101 | 23.19 |
| ST-Frame without positional encoding | ResNet101 | 21.95 |
| LT-Frame with positional encoding | ResNet101 | 22.86 |

Table 4: Ablation study of spatial offsets and positional encoding (%).

are included, the improvement in the precision ($AP_s^{@.5}$) of uniform sampling is minimal, as its smaller time window fails to capture extensive contextual information. Random sampling's $AP_s^{@.5}$ improves, and although still random, a larger temporal window permits it to sample a greater variety of changing scene information. Progressive random sampling shows the best performance, efficiently extracting dense and coherent motion information from short-term frames and capturing sparse yet rich scene changes in long-term frames. Using only short-term frames, it scores 22.17%, which increases to 23.19% with the inclusion of both frame types, demonstrating its superior ability to leverage temporal information for enhancing the detection of small objects.

**Spatio-temporal alignment encoder** In this module, comparative experiments were conducted to assess the computation of spatial offsets, as detailed in Table 4. Additionally, Figure 4 depicts the feature map representations before and after alignment aggregation, along with the sampling points on adjacent frames. This approach not only provides a quantitative analysis of the effects of spatial offset calculation but also provides qualitative insight into the impact on feature map alignment across different frames, thereby enhancing our understanding of the encoder's performance in handling temporal and spatial variations.

According to Table 4, employing convolutional offsets for spatial alignment is essential, as it mitigates the impact of spatial variations between adjacent frames, thereby accurately and effectively enhancing the pixel-level feature representation of the current frame.

**Long short-term feature aggregation module** In this module, positional encoding serves as an important component by providing additional relative positioning information between objects in short-term frames while potentially introducing misleading information into long-term frames. To investigate the specific impact of positional encoding on the detection of small objects, a series of comparative experiments were designed. The experiments first examined short-term frames without positional encoding, followed by short-term and long-term frames with positional encoding, with results presented in Table 4. According to Table 4, the $AP_s^{@.5}$ for short-term frames with positional encoding is 1.24% higher than those without, confirming the effectiveness of positional encoding in improving the model's ability to capture dynamic spatial relationships, particularly for fast-moving or small-sized objects. This improvement comes mainly from the additional spatial information provided by positional encoding, allowing the model to interpret dynamic spatial changes more accurately and make better predictions.

However, the $AP_s^{@.5}$ for long-term frames with positional encoding showed a slight decrease (from 23.19% to 22.86%) due to the less direct spatial relationships between long-term frames and the current frame. Long-term frames often contain larger scene changes and background information, and their spatial continuity may be disrupted by temporal distance, potentially introducing irrelevant or misleading spatial details, which can hinder accurate feature interpretation.

## 5 CONCLUSION

This paper proposes a long short-term transformer model that leverages temporal information to enhance features and improve small object detection in videos. A spatio-temporal alignment encoder is introduced to align features between adjacent frames, eliminating spatial variations and enhancing pixel-level features using deformable attention. To capture dense motion in short-term frames and global context in long-term frames, a progressive random sampling strategy is used to densely sample short-term frames and sparsely sample long-term frames. A long short-term module then incrementally aggregates scene information from long-term frames and motion and appearance details from short-term frames into the current frame, improving small object representation. Comparative and ablation studies on the Visdrones2019-VID and UAVDT datasets show that our method outperforms state-of-the-art approaches in detecting small objects in videos.

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
