# OpenReview forum: "LSTT:LONG SHORT-TERM TRANSFORMER FOR VIDEO SMALL OBJECT DETECTION"
_ICLR.cc/2025/Conference — ICLR 2025 Conference Withdrawn Submission_

### Official Review · Reviewer_SeGM · 2024-10-27

**Soundness:** 2
**Presentation:** 2
**Contribution:** 2
**Rating:** 3
**Confidence:** 4

**Summary:**

The paper focuses on improving the performance of small object detection for videos. To address this issue, the authors proposed an End-to-End framework called LSTT(Long Short Term Transformer). It contains a spatial-temporal alignment encoder to align spatial features across frames based on their temporal distribution and a Long Short term Feature Aggregation module to aggregate different types of queries. These queries dynamically fuse the short term spatial-temporal information with the long term spatial-temporal information. Moreover, the authors developed a sampling strategy that allows densely sampling frames close to the current frame and sparsely sampling frames far from the current frame.

**Strengths:**

1) The proposed sampling strategy is inspiring, which provides a new way to process video frames based on frame correlation compared to uniform sampling and random sampling.
2) Employing deformable transformer and spatial offset to align spatial features for short-term frames addresses the blurring problem caused by motion.

**Weaknesses:**

1) Some notations lack explanations and may cause confusion. For example, $\phi$ in Eq.2, the $dim$ in Eq.3, "DTE" in Figure 4.
2) The compared methods seem to be out-of-date, while only two methods are after 2020. Newer methods should be concluded because different datasets are used to test the performance. The ablation study for other parts of the model is also insufficient, and more components are to be examined.

**Questions:**

1) Compared to uniform sampling and random sampling, SlowFast[Ref. 1] samples frames in two different frame rates. Ablation study for exponential sampling proposed in the paper and sampling in two rates for short-term and long-term frames should be added.
2) Are there any ablation study on the proposed long short-term feature aggregation module? Only sampling strategy, spatial offsets and positional encoding in aggregation module were tested.
3) What about the differences and the advantages compared to the Lstfe-net[Ref. 2], especially the spatial-temporal feature alignment module in Lstfe-net[Ref. 2] and the spatial temporal alignment encoder in the paper because these parts are similar. Also please discuss about the differences and the advantages for the proposed aggregation module compared to the one in Lstfe-net.
4) Where is the rationale of dynamic query generation? On Line 136-139, it claims that "This module utilizes a dynamic query generation method" but in Figure 1 the dynamic query generation become an independent part without any explanation.

[Ref. 1] Feichtenhofer, H. Fan, J. Malik, K. He, Slowfast networks for video recognition, in: Proceedings of the IEEE International Conference on Computer Vision (ICCV), 2019, pp. 6201–6210.

[Ref. 2] Xiao, Y. Wu, Y. Chen, S. Wang, Z. Wang, J. Ma, Lstfe-net: Long short-term feature enhancement network for video small object detection, in: Proceedings of the IEEE Conference on Computer Vision and Pattern Recognition (CVPR), 2023, pp. 14613–14622.

---

### Official Review · Reviewer_rk6h · 2024-10-31

**Soundness:** 2
**Presentation:** 2
**Contribution:** 2
**Rating:** 5
**Confidence:** 4

**Summary:**

This paper introduces a long short-term transformer network  designed for small object detection in videos. Long-term frames capture global contextual information, enhancing the model's ability to represent background scenes, and short-term frames provides dynamic information closely related to current detection frame. Extensive experiments on the VisDrone-VID and UAVDT datasets demonstrate the effectiveness of this paper.

**Strengths:**

1. Detecting small objects is of paramount importance in practical, real-world applications.

2. Incorporating long short-term feature modeling appears to be a rational approach.

**Weaknesses:**

1. Regarding small object detection, I have reservations about relying solely on the mAP metric. I believe that comparing precision under a fixed recall threshold would provide a more accurate assessment.

2. Could you provide a comparison with the well-established ImageNet-VID benchmark to demonstrate the scalability of the proposed method?

3. I feel that the proposed technical contribution section still lacks innovative insights. As we are both aware, long-term feature modeling is beneficial for video object detection, as evidenced by studies such as [1] and [2].



[1] Jiang et al., "Learning where to focus for efficient video object detection," ECCV 2020.

[2] Jiang et al., "Video object detection with locally-weighted deformable neighbors," AAAI 2019.

**Questions:**

My primary concerns can be distilled into two key points:

1. The first pertains to the benchmarking of small object detection, specifically by evaluating precision at a fixed recall level.

2. The second involves a discussion on the differences in spatial-temporal feature aggregation in relation to the aforementioned references.

---

### Official Review · Reviewer_xocD · 2024-11-03

**Soundness:** 2
**Presentation:** 1
**Contribution:** 1
**Rating:** 3
**Confidence:** 5

**Summary:**

This paper presents a Long Short-Term Transformer (LSTT) network aimed at improving small object detection in video sequences. The proposed approach leverages both long-term and short-term frames: long-term frames capture broader contextual information, enhancing background scene representation, while short-term frames focus on more immediate, dynamic information associated with the current detection target.

**Strengths:**

The paper offers a structured approach to tackling the challenge of small object detection in video sequences, specifically targeting issues in capturing sufficient feature representations across different frames. Its main strengths are as follows:
1.Novel Sampling Strategy: A novel sampling strategy is introduced, which balances the extraction of global contextual information from long-term frames with the detailed motion and appearance features from short-term frames.
2.Structured Problem Addressing: The paper clearly identifies the limitations in traditional methods for small object detection in videos, such as uniform aggregation and lack of targeted spatio-temporal modeling, and addresses these issues with a targeted approach.

**Weaknesses:**

1.Limited Novelty in Approach: The core idea of combining long-term and short-term features for contextual and dynamic information extraction is a well-known concept in the field, and the paper lacks sufficient novelty in this regard. Many existing works have explored similar strategies for feature extraction and temporal aggregation.

2.Need for Visual Analysis: It would be beneficial to provide visual analyses, particularly focusing on the "Feature Extraction Information Extraction Long Short-term Feature Aggregation" process to validate the effectiveness of the proposed approach more comprehensively.

3.Code Availability: Providing the code, or at least core modules, would be very helpful for reviewers and researchers to evaluate and understand the contributions more fully. This would also enhance reproducibility and transparency.

**Questions:**

1.In Figure 1, how does the model ensure that Feature Extraction and Information Extraction separately capture features and information without overlap or redundancy?
2.How does the model specifically differentiate between long-term and short-term features to ensure targeted modeling, and how are these features fused in the later stages?

---

### Official Review · Reviewer_4Qpj · 2024-11-04

**Soundness:** 1
**Presentation:** 1
**Contribution:** 1
**Rating:** 3
**Confidence:** 3

**Summary:**

This paper presents the Long Short-Term Transformer (LSTT) for small object detection in videos. LSTT proposes a progressive random sampling strategy and employs a spatio-temporal alignment encoder to align features between adjacent frames. Additionally, it introduces a long short-term feature aggregation module to integrate scene information from long-term frames as well as motion and appearance details from short-term frames into the current frame.

**Strengths:**

1. The proposed framework is simple and easy to understand.
2. The proposed method achieves new state-of-the-art performance on the Visdrones2019-VID and UAVDT datasets.

**Weaknesses:**

1. The paper is poorly organized, lacking clear explanations for symbols and notations, and the overall writing needs improvement.
2. This work lacks novelty, primarily relying on structures from previous works (e.g., deformable attention) to address small object detection in videos.
3. The proposed LSTT method includes numerous modules, which could potentially result in an excessive number of parameters and high computational complexity.

**Questions:**

Could the authors provide a comparison of the number of parameters and computational complexity in Tables 1 and 2?

---

### Note · Authors · 2024-11-13

I have read and agree with the venue's withdrawal policy on behalf of myself and my co-authors.